# Predicting heterogeneous ice nucleation with a data-driven approach

Martin Fitzner [1], Philipp Pedevilla[1] & Angelos Michaelides [1,2]✉

Water in nature predominantly freezes with the help of foreign materials through a process known as heterogeneous ice nucleation. Although this effect was exploited more than seven decades ago in Vonnegut's pioneering cloud seeding experiments, it remains unclear what makes a material a good ice former. Here, we show through a machine learning analysis of nucleation simulations on a database of diverse model substrates that a set of physical descriptors for heterogeneous ice nucleation can be identified. Our results reveal that, beyond Vonnegut's connection with the lattice match to ice, three new microscopic factors help to predict the ice nucleating ability. These are: local ordering induced in liquid water, density reduction of liquid water near the surface and corrugation of the adsorption energy landscape felt by water. With this we take a step towards quantitative understanding of heterogeneous ice nucleation and the in silico design of materials to control ice formation.

[1] Thomas Young Centre, London Centre for Nanotechnology and Department of Physics and Astronomy, University College London, Gower Street, London WC1E 6BT, UK. [2] Department of Chemistry, University of Cambridge, Lensfield Road, Cambridge CB2 1EW, UK. ✉email: angelos.michaelides@ucl.ac.uk

There are few, if any, physical processes as ubiquitous as the freezing of water. It is of fundamental importance to the environment, technology, and biology. Generally when water freezes the initial nucleation event takes place at the surface of some foreign material. A remarkably broad variety of materials can act as nucleating agents, ranging from minerals[1], to carbonaceous materials[2], to organic molecules and organic matter[3]. However, it is still not understood why some materials are better than others when it comes to promoting ice formation[4]. Stated differently, robust connections between the physiochemical properties of materials and their potency as ice nucleating agents have yet to be established.

Attempts to understand this issue go back to at least the pioneering experiments of Vonnegut in the late 1940s and early 1950s[5–7]. Working with the hypothesis that effective ice nucleating agents should offer a template for ice, Vonnegut identified AgI as one such material. Specifically, he noted that the basal faces of AgI and ice I$_h$ have lattice constants that fall within 1.5% of each other. Various studies have now shown that AgI is an effective ice nucleating agent and it is widely used to this day to seed clouds. However, counter-examples exist of materials that have an equally small lattice mismatch with ice and yet are ineffective ice nucleating agents, BaF$_2$ being the most widely studied of these[8–10]. Thus, the lattice match alone is not a reliable or robust descriptor for ice nucleation ability. Beyond the lattice match, in the seventies Pruppacher and Klett[11] introduced a set of requirements for effective ice nucleating agents. While some of these deal with macroscopic properties from an atmospheric chemistry viewpoint (insolubility and aerosol size requirement) the others (chemical bond, active site and Vonnegut's crystallographic match) deal with microscopic characteristics. Pruppacher and Klett's requirements are widely discussed. However, there are again many known exceptions to them and they do not easily lend themselves to quantitative comparisons between materials[12–14].

The last decade or so has been a resurgence of interest in understanding and obtaining well-defined molecular-level information of heterogeneous ice nucleation. From an experimental point of view insights into ice formation on specific substrates have been obtained. These have included measurements that reveal atomic-level structural information on well-defined substrates[15–20] as well as measurements of ice nucleation on materials of atmospheric relevance[1,3,13,21–23], and more[24–27]. Computer simulations have also proven to be a powerful tool in providing insights into heterogeneous[28–33] (and homogeneous[34–38]) ice nucleation, particularly at the atomic level. There have also been systematic trend studies focused on understanding the connections between nucleation rate or temperature with the lattice constant of the substrate, surface hydrophilicity, hydroxyl group structure and symmetry, and surface flexibility[39–47]. Various other influential surface characteristics such as charge-distribution[48] or nanotexture[49] have also been explored.

While these studies have significantly deepened understanding of heterogeneous ice nucleation, they have not led to a robust set of descriptors that can predict the ice nucleating (IN) ability for a diverse set of substrates. Even the question of how many different properties of a surface are relevant to ice nucleation is unclear. Building on the earlier work, methodological developments that enable the high throughput computational study of heterogeneous ice nucleation[41,47,50,51], and machine-learning (ML) approaches for the analysis of large data-sets, the prospect of understanding such complex relations is now in reach. This is the approach we take in the current study through the development of predictive ML models trained on molecular dynamics (MD) simulations of heterogeneous ice nucleation and random-structure searches of adsorbed water clusters and overlayers. From this, we identify four key descriptors that when used

together can accurately predict the ice nucleating potency of solid substrates. This study deepens our understanding of what makes materials efficient ice nucleating agents and is a step toward a comprehensive and predictive set of descriptors which can be used in future screenings and to guide and interpret experiments.

## Results

**Ice nucleation simulations.** As a first step to identify the important factors for predicting the IN ability of substrates, we need to have suitable data. We acquire these by performing MD simulations of supercooled water in contact with a large set of structurally diverse model substrates. Our dataset comprises the 400 Lennard–Jones substrates of ref. [41], the OH-group patterns of ref. [47], graphitic surfaces with modulated water–substrate interaction strength similar to those used in ref. [43], graphene oxide as modeled in ref. [40], and several additional OH-group patterns with different symmetries from those reported in ref. [47]. In total we have 900 substrates. On each system we perform cooling ramps and establish the temperature at which the liquid freezes, termed nucleation temperature $T_n$. Additional information on the simulations is given in the methods and Supplementary Note 1.

In this study we have used the mW model to represent water[50]. This has proven to be an extremely powerful model for understanding water and ice, see e.g., refs. [29,34,35,39,51]. Its computational efficiency means that we have been able to screen 900 substrates for their nucleation ability. Such a broad study is barely conceivable with an atomistic model because of the computational cost of performing nucleation simulations with such models[52,53]. Thus, using an atomistic model would have severely limited the breadth of our study to only a small subset of substrates or to substrates with a very ice-like geometry such as AgI[54]. From such a study we would not have sufficient data diversity to identify robust descriptors for $T_n$. Thus, at present, a coarse-grained model such as mW is most appropriate for a broad screening study such as the current one. However, it is important to emphasize that mW does not capture effects such as polarization, water dissociation or hydrogen bond asymmetry. Understanding how these and other effects influence the conclusions reached in the current study—through simulations with more sophisticated water models—will make interesting work for the future.

As a first step to analyze our data we looked to see if the traditional lattice match descriptor could be used to explain the nucleation temperatures obtained. As an additional potential descriptor we also considered the water–monomer adsorption energy on the surfaces; this is considered a proxy for the hydrophobicity/hyrophilicity of the surface[17,43,44,55]. The scatterplot shown in Fig. 1a summarizes the distinction between good and bad nucleating substrates in our database. As a guide to the eye we define good nucleating agents with a relatively high nucleation temperature ($T_n > 225$ K) and substrates that are bad with $T_n \leq 225$ K. Although there is a slight preponderance of good nucleating substrates in the small lattice mismatch regime, the correlation is clearly not good and there are many substrates which are effective ice nucleating agents and are poorly lattice matched to ice. In terms of the water–surface interaction strength no obvious connection with ice nucleating ability is found in the data. All other simple linear descriptors considered suffered similar problems and none was found to be an accurate descriptor for heterogeneous ice nucleation.

**Nucleation temperature prediction.** To help identify descriptors and obtain quantitative insight into heterogeneous ice nucleation we turned to a data-driven methodology. As summarized in Fig. 2, our approach involves several steps to select the important

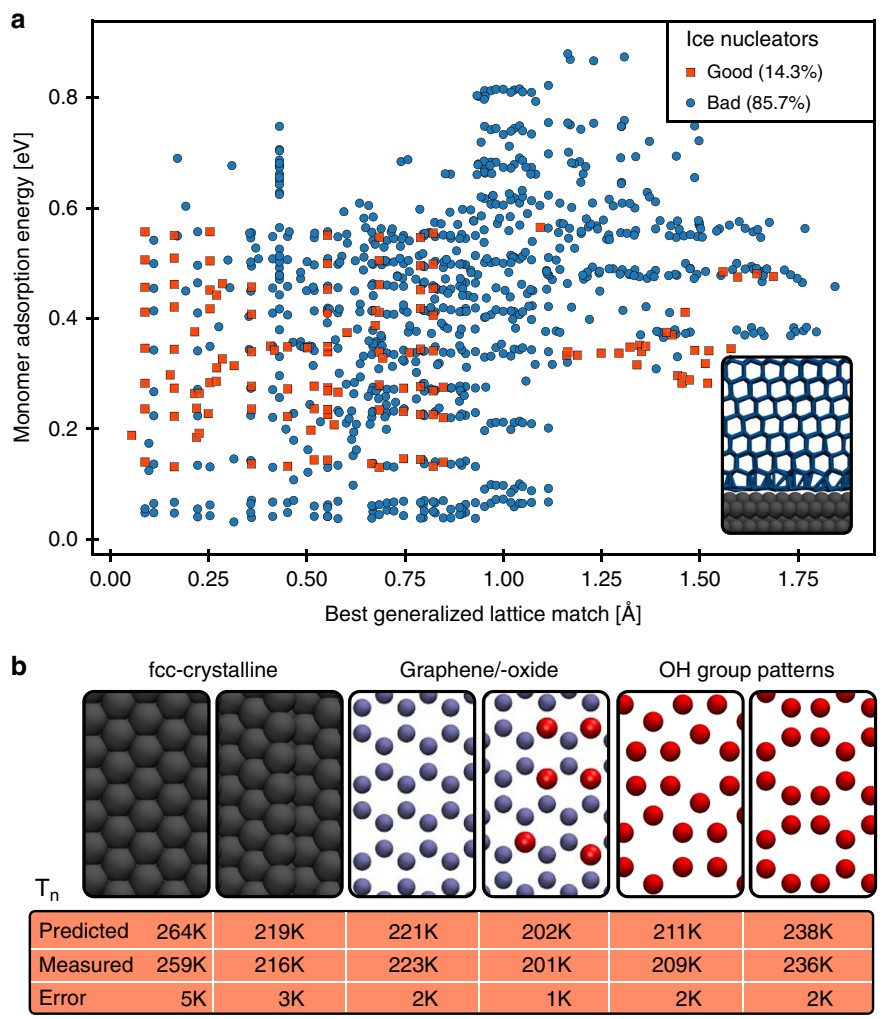

**Fig. 1 Overview of the database examined and the performance of our ML model. a** Scatterplot of monomer adsorption energy versus the best generalized lattice match for all substrates. Zero corresponds to a perfect lattice match between ice and the substrate. Substrates are classified as good ice nucleators if their nucleation temperature was above 225 K and as bad otherwise. The inset on the bottom right shows a side view of a typical simulation cell after the nucleation event. Source data are provided as a Source data file. **b** Exemplary substrates studied in this work (top down view): fcc-crystalline substrates with Lennard–Jones interaction (gray), graphitic surfaces (light blue) and OH-group patterns (red). The table below indicates measured nucleation temperatures and the prediction of our best ML model (when the respective substrate was outside of the training set).

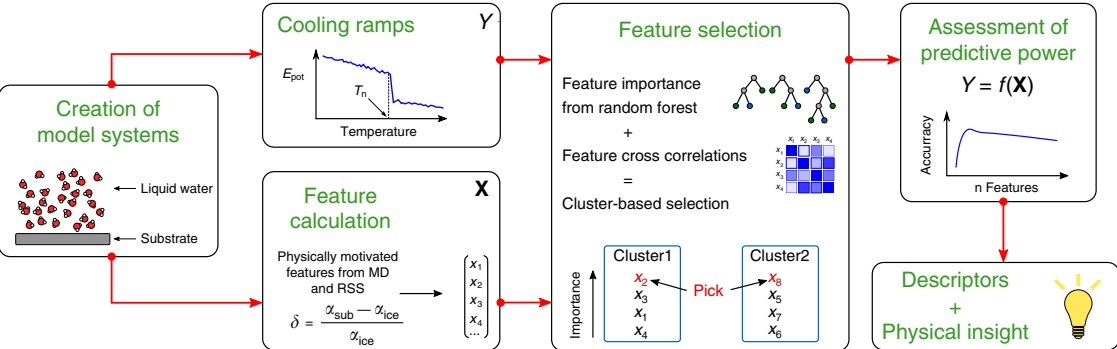

**Fig. 2 Overview of the data-driven workflow employed in this study.** In brief, we start by creating a diverse set of substrates covered by liquid water. By performing MD cooling ramps we establish their nucleation temperature $T_n$, which is later used as dependent variable $Y$. Separately, we use these model systems to compute a large set of features, to be tested as independent variables $X$. We feed these data to a feature selection approach to select the most important parts of $X$. This is done by using the feature cross-correlations for distance-based clustering and then selecting from each cluster the most important feature, where the importance comes from a random forest model using all features. Subsequently, we check the predictive power over $Y$ by machine-learning the dependence $Y = f(X)$. Last, we employ state-of-the art feature interpretation techniques to gain physical understanding of the selected features.

features for predicting $T_n$. After having measured $T_n$, a large set of physically motivated features (e.g., the liquid density, number of nearest neighbors, adsorption energies etc.) are calculated from separate simulations. We feed these data to a cluster-based feature selection approach. This reduces the number of features used in the model to only a handful, which enables a more in-depth analysis later on. We subsequently check their predictive power over $T_n$. Last, we employ state-of-the art feature interpretation techniques to gain physical understanding of the selected features. In the following we will call those selected features descriptors to distinguish them from the other calculated quantities. A more detailed overview of the approach is given in the Methods, Supplementary Note 3 and Supplementary Figs. 3 and 4. We note here that through our cooling protocol for establishing $T_n$ from repeated cooling ramps on each system we have a natural uncertainty on $T_n$ of ca. 6 K (±3 K, where 3 K is the average standard deviation of $T_n$ for all substrates). This represents a lower bound for the best possible prediction (in terms of the root mean square error, RMSE) we can achieve.

We have already seen that analysis of the data based on linear descriptors is unsatisfactory. This is confirmed by testing a linear baseline model on our data (an elastic net model which is a version of linear regression). As shown in Fig. 3a this baseline model is rather insensitive to the number of descriptors considered and yields a best RMSE of 12.5 K on predictions of $T_n$ across our entire database. The expected error window can be approximated by ± the RMSE. Given that the range of nucleation temperatures observed in this work spans 70 K, this is mediocre performance at best. Indeed, since the typical experimental range of ice nucleation temperatures explored is about 35 K[21], a prediction window of 25 K (±12.5 K) is not sufficiently accurate to make reliable quantitative predictions of the ice nucleating ability of substrates.

Turning now toward the performance of the best ML model identified in this study, we find that it performs well with RMSE values approaching 6 K (Fig. 3a). This yields a prediction window of 12 K (±6 K), half that of the baseline model, and sufficiently narrow to be considered a good prediction of $T_n$. With this accuracy, substrates could be classified into several classes of nucleators to reliably distinguish good from bad ice forming substrates. We are not aware of any previous works that predicted the IN ability across a diversity of many different (out-of-training-set) substrates: The closest previous studies[47,56] focused on more qualitative insight, and did not report quantitative metrics. A representative comparison of predicted and measured $T_n$ for the best ML model is shown in the inset of Fig. 3a where a $R^2$ value of 0.856 indicates a good correlation between predicted and measured values. A common interpretation of this metric is that the model is able to explain ~86% of the variance of $T_n$. We have also used dimensionality reduction to see whether good and bad nucleators are visually separated, which is indeed the case for our best ML model (see Supplementary Note 7 and Supplementary Fig. 8). This provides further evidence that we have identified meaningful descriptors that can distinguish good from bad ice nucleators.

Another important observation to make from Fig. 3a is that there are four main contributors to predicting $T_n$, since the model performance does not improve significantly after including more than four descriptors. The first and most important descriptor found is the lattice match between the substrate and ice. Given previous work, this is not unexpected[5,41,57]. However, its identification in a purely data-driven manner is a verification of the efficacy of the approach developed here. The other descriptors are essentially measures of the following physical properties: (i) the local ordering in liquid water; (ii) the local density reduction

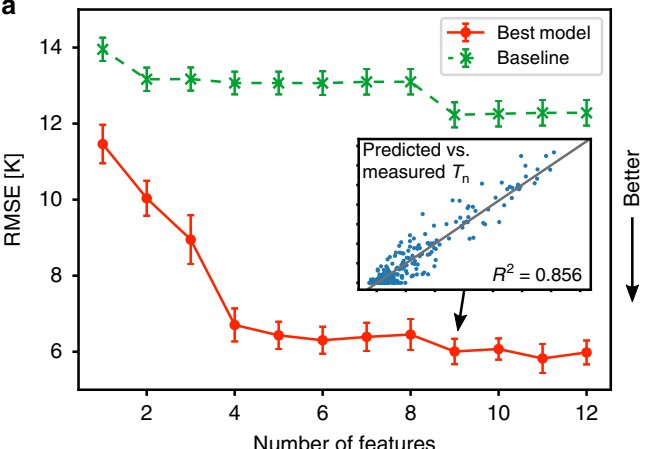

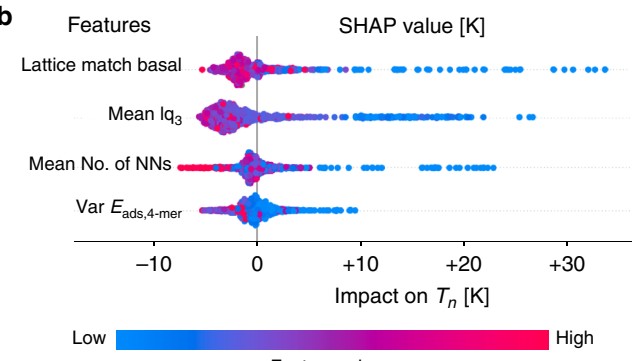

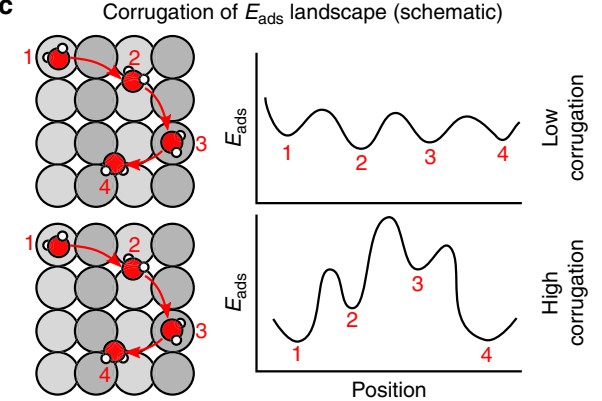

**Fig. 3 Model performance and descriptor identification. a** Performance of the linear baseline model and the best ML model identified in this study as a function of the number of included descriptors. The inset shows the correlation of predicted and measured $T_n$ for the ML model using nine descriptors. Error bars represent the standard deviation over 20 independent train-test splits. Source data are provided as a Source data file. **b** SHAP values (model impact) for the first four included descriptors from panel (**a**). These are (i) the lattice match to the basal face of ice; (ii) the mean $lq_3$ (as defined in the text); (iii) the mean number of nearest neighbors within 3.4 Å; and (iv) the variance of water tetramer adsorption energies. The color scale goes from the lowest (blue) to the highest (red) value present in our dataset for each descriptor, respectively. **c** Sketch illustrating substrates (top view on the left) that are structurally similar. The water–monomer adsorption energies ($E_{ads}$ shown schematically on the right) for different positions are not very different in the case on the top substrate while in the bottom substrate they are very different, leading to a more corrugated adsorption energy landscape.

in the liquid near the surface; and (iii) the corrugation of the adsorption energy landscape felt by water. When these additional descriptors are taken into consideration, the prediction error is reduced by half, compared to just lattice match-based predictions alone. This result is encouraging and suggests that we can find the missing contributions that help to predict whether a substrate is good or bad at ice nucleation. We note that although different feature selection methods can yield slightly different results (see Supplementary Note 4 and Supplementary Fig. 5), the descriptors identified always came from these broad physical families, indicating that our findings are robust. In the following sections we discuss all four of the key descriptors identified in detail.

**Physical interpretation of descriptors**. We have established that we can build predictive models (with a low number of descriptors) for the IN ability of different substrates. To extract physical insight and guide our interpretation of the black-box decision of the ML model we apply SHapley Additive exPlanations (SHAP)[58]. SHAP values come from a step-wise decomposition of the model decision rooted in game theory, by essentially comparing the outcome difference of including or not including a certain feature. This means that a feature impact of, e.g., +30 for the lattice match feature means that the model's prediction for $T_n$ was increased by 30 K due to that feature. SHAP values are always for a specific data point (i.e., in our case substrate) and must be regarded as a distribution per feature.

In Fig. 3b, we display the SHAP value distribution for the first four descriptors included in Fig. 3a. Each colored point corresponds to a substrate, while its color relates to the values (low or high) of the corresponding descriptor for that row (the color is not to be confused with high or low $T_n$). Whether a point is to the right or left of the graph indicates whether for this substrate the descriptor of that row caused a positive or negative impact on the value of the predicted $T_n$. We now discuss what we have learned from the SHAP analysis and the four key descriptors that have been identified.

**Lattice matches**. The most familiar descriptor that proves to be important in predicting $T_n$ is the lattice match of the substrate with the basal face of ice, specifically the 2-dimensional in-plane lattice match. If we examine the SHAP values for this descriptor (first row in Fig. 3b) we can see that low values (blue dots, corresponding to a good match) often contribute to predicting a much higher $T_n$. This is in agreement with previous results[41,56,57] which show that a good lattice match to the basal face can result in a good IN substrate. However, as noted already, we find many exceptions where a good lattice match to ice does not lead to an efficient ice nucleating substrate.

To capture also the match to other ice faces we calculated a generalized lattice match $\zeta$:

$$\zeta = \min_{\mathbf{r}_0, \theta} \left( \sqrt{\frac{1}{N_M} \sum_{i=1}^{N_{ice}} \left( \mathbf{r}_i(\mathbf{r}_0, \theta) - \mathbf{r}_s \right)^2} \right) \quad (1)$$

Here, $\mathbf{r}_0$ is the position vector of the center of mass of a randomly displaced ice crystallite (corresponding to a certain ice face), $\theta$ is its rotational orientation relative to the surface normal, $N_{ice}$ is the number of water molecules in the ice crystallite, $\mathbf{r}_i(\mathbf{r}_0, \theta)$ is the position of oxygen atom $i$ at a given $\mathbf{r}_0$ and $\theta$ and $\mathbf{r}_s$ are the closest substrate atom to water molecule $i$. If water molecule $i$ does not have a substrate atom close by (threshold: 3.3 Å), then this water molecule is omitted from the calculation. By choosing different ice crystallites as reference we can probe the match to different faces of ice. Here, we have focused on the basal, prism and secondary prism faces of $I_h$ and the (111) and (001) faces of $I_c$. The results for the generalized lattice match have been discussed

previously and are shown in Fig. 1a. Upon considering the relative importance of the different faces in predicting IN ability we find that the lattice match to the basal face is the most important, with the match to the prism face being the next most important.

**Local ordering**. We find that the local order of the liquid is another important descriptor of a good ice nucleating substrate. Local order can be conveniently defined as:

$$q_{lm}(i) = \frac{1}{N_b(i)} \sum_{k=1}^{N_b(i)} Y_{lm}(\theta_{ik}, \phi_{ik}) \quad (2)$$

where $Y_{lm}$ are spherical harmonics and $\theta_{ik}$ and $\phi_{ik}$ are the relative orientational angles between the molecule $i$ and $k$. The sum goes over the $N_b(i)$ neighbors of molecule $i$ (i.e., within 3.4 Å). For a given $l$ we then compute the quantity for all possible values of $m$ and store them in a vector $\overrightarrow{q}_l(i)$ containing $2l + 1$ components. From this we calculate values $lq_l$ according to:

$$lq_l(i) = \frac{1}{N_b(i)} \sum_{k=1}^{N_b(i)} \frac{\overrightarrow{q}_l(i) \cdot \overrightarrow{q}_l(k)}{\left| \overrightarrow{q}_l(i) \right| \cdot \left| \overrightarrow{q}_l(k) \right|} \quad (3)$$

Of the various local order parameters evaluated (see Supplementary Note 2 and Supplementary Fig. 2), we find that $lq_3$ is a particularly good descriptor. For $lq_3$, lower values indicate a tetrahedral environment, and as shown in Fig. 3b (second row), substrates with a low mean $lq_3$ are predicted as good nucleators while mixed and higher values tend to be associated with poor nucleators.

Further, in addition to the mean, the standard deviation of $lq_3$ values appears as the sixth selected descriptor, with higher values being favorable for nucleation. We can interpret this in a practical sense: Since these statistics come from MD trajectories, an increased variability of $lq_3$ indicates a higher fluctuation of ordered structures due to the influence of the substrate. This is consistent with our earlier suggestions that pre-critical fluctuations[28] are indicative of the IN ability as well as the type of ice that will form. Thus, the variability of local ordering can be seen as a measure of pre-critical fluctuations, an aspect that could be studied by time-resolved experiments.

We note that we have calculated the various order parameters from MD trajectories at the coexistence temperature for all substrates. This is interesting because the order within the liquid at the coexistence temperature can readily be computed with more sophisticated water and substrate models than those used here and it can also be established experimentally with techniques such as surface X-ray diffraction[59].

**Liquid density reduction**. The third descriptor appearing in Fig. 3b is the average number of nearest neighbors (third row) that we count within a sphere of radius 3.4 Å. This descriptor is connected to the density of the liquid. We find that substrates that are able to decrease the number of nearest neighbors (i.e., the local density, blue points) due to structuring of the liquid are on average better nucleators. This has a straightforward physical interpretation: If the average density is higher in the liquid than in the crystal (which is true for water), then upon nucleating the emerging crystal has to push out the excess molecules in its vicinity, and this likely contributes to the nucleation barrier. If a surface is able to reduce the liquid density in its proximity, this will therefore benefit this regions' ability to nucleate ice.

We are not aware of previous discussions of this microscopic density principle for heterogeneous ice nucleation, despite its straightforward physical basis. For homogeneous nucleation, Li and coworkers[34] observed a similar trend with the density,

presumably for the same physical reason. In addition, our result shows that the density reduction is also a vital component of good heterogeneous nucleators and could thus potentially be used to control heterogeneous nucleation.

From all the substrates considered, we find substrates that both decrease and increase the average number of neighbors in the adjacent liquid. This means that the density descriptor could also be used to screen for both nucleation-promoting and nucleation-inhibiting (these may be overly dense at the interface) surfaces.

**Corrugation of the adsorption energy landscape**. Finally, we find that adsorption energies ($E_{ads}$) play an important role as descriptors. However, unexpectedly it is not absolute values, but rather the diversity of possible adsorption energy values that is important. We computed these descriptors by performing a large set of random-structure searches for diverse ice-like structures on each substrate to characterize their $E_{ads}$ landscape. The statistics that appear as important (e.g., for water tetramers in Fig. 3b) are measures of the spread of $E_{ads}$ values (such as variance, range or (Shannon-)entropy) and not absolute values (such as mean or median). We interpret this as the corrugation of the adsorption energy landscape being measured, with a less rugged environment (smaller variance) being more beneficial to IN. In Fig. 3c, we show an illustrative sketch of different adsorption energy landscapes, depending on the variability of the adsorption strengths of the different sites. It is important to realize that this measures not the absolute (average) adsorption strength but rather the diversity of the $E_{ads}$ distribution.

Another unexpected finding is that it is not the most ice-similar structures that are the most relevant structures for this measure. We see that the adsorption uniformity for tetramers and pentamers is most important, followed by monomers and dimers. The hexamers or cages investigated received a lower importance rating. This perhaps indicates that interfacial structures that are not necessarily ice-like play an important role and a good ice nucleator does not require the interfacial water layers to be precisely ice-like. The potential role of interfacial structures to the nucleation process has been noted. For instance a defect structure with 5-fold symmetry was observed during homogeneous nucleation[60]. We have not directly observed any particular nucleating structures in this work, but the connection to $E_{ads}$ of pentamers is possible.

**Remarks on non-selected features**. After having discussed the most important descriptor types for predicting $T_n$, it is interesting to consider the features that have not been selected by our algorithm. First, we note that if a feature has not been selected it does not mean that it is entirely unimportant. This is because our approach takes into consideration both the feature importance and the correlation with already selected features, so a feature not being selected could also come from redundancy with other descriptors. In practice it will be a combination of both effects. Second, the number of non-selected features vastly outnumbers the ones that are selected (for a listing see Supplementary Note 2), so in the following we focus on a few examples that are surprising or are related to previous studies.

We have also computed the layering as defined in ref. [39] and refs. [43,44], where the absolute deviation from bulk number density is measured and integrated, treating increased and decreased densities the same. Our results indicate that rather than an absolute change in density, a density reduction is desirable, hence making a measurement for this the better descriptor. Also, a certain correlation between the layering and density reduction is expected, which might also contribute to layering not being selected.

Most work on heterogeneous nucleation in general focuses on structural aspects of the substrate or the interfacial water. Recent work, however, has highlighted the role of heterogeneous dynamics for supercooled water in the homogeneous case[36]. Despite this, the dynamics near liquid interface remain largely unexplored. To have a simple measure for liquid dynamics near the interface we computed mean displacements at timescales ranging from 1 to 150 ps. A distinction between different layers perpendicular to the surface was also made. None of these features has been selected in the end, which is somewhat of a surprise. A possible explanation is the following: Most of our substrates will decrease the mobility of interfacial water molecules, thus measures of displacements will offer little distinction between them. For hydrophobic surfaces this would not be the case, displacements possibly being higher. However, for those substrates then also trivially other descriptors would indicate the trend with $T_n$, such as the local ordering, hence making displacements unhelpful.

A large group of features not appearing useful are velocities and forces (distinguished in layers and perpendicular/parallel to the surface normal). These are expected to differ from bulk near the interface, potentially carrying information about $T_n$. However, this was not the case.

## Discussion

In this work we started by establishing the IN ability of a large number of diverse substrates in contact with water. We then developed a data-driven approach based on machine-learning methods to identify descriptors for heterogeneous ice nucleation and established that the resulting models are indeed predictive. A key conclusion is that no single descriptor has been identified that can reasonably well predict the IN ability. For each of the important descriptors there are exceptions where they are not predictive if used on their own. However, by using the following four microscopic principles in conjunction, we can reliably predict the IN ability of a given substrate. They are:

- Lattice match of the substrate to a low-index face of ice;
- Local tetrahedral ordering of the liquid near the surface, ideally measured with lq$_3$, as well as pre-critical fluctuations (measured as the variability of lq$_3$);
- Density reduction of the liquid near the surface;
- Corrugation of the adsorption energy landscape, with a smooth energy landscape preferred to a rugged one.

With this we have shown that a ML model can be developed that is capable of learning the important contributions for a quantitative prediction of heterogeneous ice nucleation. This has revealed the particular characteristics beyond the lattice match that help to make a surface a good or bad ice nucleator. While the rediscovery of the importance of lattice match is not new, the fact that it falls out of the ML model serves as a verification of our method. Of the three other descriptors identified, local ordering has received limited prior attention, and the descriptors on the corrugation of the adsorption energy landscape and the density reduction near the surface are new concepts in heterogeneous ice nucleation.

Two out of the four descriptors we found are more directly related to intrinsic surface properties (lattice match and adsorption energy landscape) while the others describe interfacial water. Overall, it is more desirable to have descriptors that relate to surface chemistry alone. This goal, however, is much more challenging and for future work we propose to split the process of understanding heterogeneous IN into two steps: (i) relate IN activity to interfacial water properties and then (ii) relate surface chemistry to interfacial water properties.

An important question is whether the descriptors identified in this study will apply if ice nucleation is explored with other simulation approaches (e.g., other water models) or in experiment. The nature of the ML model means that there is unlikely to be a numerical one-to-one correspondence between the descriptors identified in this study and those obtained with other water models. In other words, a covariate shift between descriptor values of different water models is to be expected. However, with this in mind, we have refrained in the current study from reporting specific threshold values for descriptors. Instead, we have reported and discussed broad categories or families of descriptors. The categories of descriptors are all clear and physically motivated quantities that can be probed and tested with other water models and with experiment. In addition, we note that the descriptors were obtained from simulations at the coexistence temperature (where ice and liquid are equally stable). Hence, there is no need to study further any system at or close to its actual (and initially unknown) freezing point. This drastically simplifies computational and experimental investigations probing the descriptors identified here, since substrates can be compared based on single temperature studies.

Before closing, we give some perspective on connections with experiments. There are a variety of experimental techniques already available to study many of the physical properties discussed in this study. The lattice match can already be probed by crystallographic methods. Density and local ordering could be probed by surface X-ray[59] or sum-frequency generation spectroscopy[61,62]. The adsorption uniformity could be studied with atomic force microscopy[63] or scanning tunneling microscopy[17,64]. Our study also calls for time-resolved methods that do not average out structural fluctuations on short timescales, to being able to investigate pre-critical fluctuations. Given our findings we are confident that any such investigation can be done at constant temperature (e.g., close to coexistence), even when two materials with drastically different IN ability are compared.

Finally, we note that this study deals with the microscopic influences on IN, i.e., essentially regarding the substrate as infinite and flat (on the order of a few atomic distances). There are other potentially influencing factors such as macroscopic hydrophobicity and large-scale roughness. In this sense, we regard our results as being of primary relevance to the so-called active sites in IN. This viewpoint can connect our work to more macroscopic studies and experiments in which the existence and location of active sites has been demonstrated[23] but their structure and composition remain elusive.

To conclude, with this work we have identified a predictive set of microscopic descriptors for the IN ability of substrates and have shown that the quantitative predictions of heterogeneous IN can be made. We expect that our findings will be useful in interpreting future ice nucleation experiments and as a guide for the development of substrates with targeted ice promoting/inhibiting abilities.

## Methods
**Systems and simulations**. We start by giving a brief overview of the workflow developed in this study. For a more detailed description the reader is referred to Supplementary Notes 3 to 8, where we discuss more aspects of the ML methods employed. A sketch describing our general approach can be found in Fig. 2. As a first step, we combine a set of model systems for heterogeneous ice nucleation both from the literature as well as newly added substrates. Our dataset comprises of 900 diverse substrates, partly inspired by refs. [40,41,43,47] as outlined in the main text. For the MD simulations, the water–water interactions are represented by the coarse-grained mW model[50], while the water–substrate interaction varies depending on the substrate type, but mostly employs variations of the water–water interaction as well as Lennard–Jones-like interactions. More simulation details can be found in Supplementary Note 1 and Supplementary Fig. 1.

On these systems we perform five cooling ramps each (at $1 \, \mathrm{K \, ns^{-1}}$) from 273 to 200 K. Cooling ramps are widely used to characterize the IN ability of a given system in simulations[39,40,47] and experiments[13,21,24]. We define as the nucleation temperature $T_n$ the temperature at which a significant drop in the potential energy ($E_{pot}$) is seen. This identifies the onset of nucleation as the drop is much more pronounced than the slope caused by the cooling. $T_n$ correlates well with IN ability since a nucleation event at a higher temperature means a larger reduction of the homogeneous nucleation barrier (which increases with higher temperature). See ref. [47] for a comparison of $T_n$ with nucleation rates. With this approach we have a measure to rank the substrates by their IN ability as well as a measure for the uncertainty in the IN ability (the standard deviation of the 5 runs). With 5 runs we strike a balance between precisely knowing $T_n$ and computational efficacy. In hindsight, we see that our best model RMSE is larger than the uncertainty from calculating $T_n$, suggesting that there is little value of establishing $T_n$ with higher precision.

**Computation of features**. To find descriptors that can predict the IN ability we first compute a large selection of features. We have generally taken into account forces, displacements, densities, generalized lattice matches, velocities, adsorption energies of different ice structures, and local structuring measures. We also distinguish different liquid water layers regarding their vicinity to the surface, and where applicable also different timescales (e.g., for average diffusion distances). These features are gathered by post-processing data from either MD or the random-structure-search approach used in ref. [65]. The various combinations amount to about 3000 initial features that will be assessed. All features we compute and the methods used to obtain them are described in more detail in Supplementary Note 2 and Supplementary Fig. 2.

**Feature ranking and selection approach**. To identify the most relevant features out of the ~3000 considered, we couple two important components. First, we calculate an importance score for each feature. This is extracted from fitting a random forest model[66] including all features. This model type has the advantage that it can easily be trained for a large number of features and comes naturally with a measure for feature importance. The measure is related to the performance of the model (on out-of-sample data points) compared to the performance after randomly permuting the values in that feature. We train the random forest to classify whether the system is a bad or good nucleator (threshold 225 K).

By construction, we expect many of the features to carry similar, if not identical, information (take for instance the average density in two adjacent layers). Thus, as a second step, we perform hierarchical clustering between the features to deal with correlations among them. As the distance $d(f_1, f_2)$ between two features $f_1$ and $f_2$ for the clustering we choose $d(f_1, f_2) = 1 - \mathrm{MIC}(f_1, f_2)$, where MIC is the maximum information criterion[67], an information-theoretical measure similar to correlation that is capable of capturing non-linear associations, which is also bound to the interval [0, 1].

We then combine these two components to pick a set of $n$ features, by forming $n$ clusters and selecting from each the feature with the highest importance (see the central part of the workflow in Fig. 2). Various approaches are used for this (see Supplementary Note 4 and Supplementary Fig. 5). Here, we show results using an average linkage approach for the clustering.

**Model performance assessment**. After selecting relevant features we check whether they are actually able to predict the IN ability of substrates in a satisfactory manner. We do this through a regression of $T_n$ and use the root mean square error (RMSE) as a performance indicator. In the Supplementary Note 5 and Supplementary Fig. 6 we also discuss a simpler classification task where we split all nucleators into good or bad ones according to Fig. 1a.

The choice of ML model presents a potential bias when assessing the predictive power of the selected features. While the feature set might contain all necessary information to predict $T_n$, a model that for instance cannot capture non-linearities might nonetheless yield bad scores. We choose as the estimator model a widely adopted variant of gradient boosting with trees, xgboost[68]. While it is beyond the scope of this study to benchmark many different ML models, we note that the accuracy assessment was repeated with random forests[66] and support vector machines[69], yielding worse results but consistent trends (these results can be found in Supplementary Note 6 and Supplementary Fig. 7).

We calculate the corresponding model score on a 30% hold-out set, repeated 20 times. The standard deviation of this is reported as the error of the score. As stratification for the splits we divide the $T_n$ range into five sub-ranges (each about 14 K). On the other 70% of the data we train the model after performing 5-fold cross-validation to select the best hyperparameters. This split is done prior to the feature selection to avoid the test set influencing the selection process. The hyperparameter search was done utilizing the Bayesian tree-structured Parzen estimator[70]. As a baseline to compare this with we use the linear elastic net model.

## Data availability
The data that support the findings of this study are published under the https://doi.org/10.6084/m9.figshare.12855563.

## Code availability

The simulation software package is open source[71], the scripts to perform random-structure search were published along with ref. [47] and the data analysis was done using open-source python packages (pandas, shap, xgboost).

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

## Acknowledgements

We are very grateful to G.C. Sosso, S.J. Cox, and B. Slater for their comments and suggestions on this manuscript. This work was supported by the European Research Council under the European Union's Seventh Framework Program (FP/2007-2013)/ERC Grant Agreement No. 616121 (HeteroIce project). We are grateful for computational resources provided by the London Centre for Nanotechnology, the UCL Grace High Performance Computing Facility (Grace@UCL), the Materials Chemistry Consortium through the EPSRC Grant No. EP/L000202 and the UK Materials and Molecular Modelling Hub for computational resources, which is partially funded by EPSRC (EP/P020194/1).

## Author contributions

M.F. and A.M. designed the research. M.F. and P.P. performed the simulations. M.F. performed the data analysis. M.F. and A.M. wrote the manuscript. All authors contributed to interpreting and analyzing the results.

## Competing interests

The authors declare no competing interests.
