## [Peer Review File · Nature Communications]

REVIEWERS' COMMENTS:

Reviewer #1 (Remarks to the Author):

This is an interesting modeling study attempting to correlate physical descriptors of a substrate with its heterogeneous ice nucleation efficiency, which is a long-standing scientific question, through machine learning approach. The authors relied on the large database produced by a computationally inexpensive water model and simplified substrate model to enable a data-driven approach for training ML model. The outcome of the study revealed four major descriptors that were shown capable of predicting heterogeneous ice nucleation temperature within a RMSE of $\pm 6\text{K}$. I find this study very well planned and executed, and the results/analysis interesting and insightful.

A main question is that out of the four identified descriptors, only one (lattice match) directly measures surface attributes, whereas other factors describe the structure and energetics of interfacial water induced by surfaces. Although I think it makes perfect physical sense that machine learning model revealed interfacial water are strongly correlated with its likelihood of turning into ice (because in general, the more similar interfacial water is to ice, the more likely it will turn into water as this can be also easily seen from a view of entropy reduction), the outcome of the study appears to benefit modeling study more than experimental study of heterogeneous ice nucleation, since local ordering and density of interfacial water are much easier to characterize in modeling than in experiment--identifying a good/bad nucleator in experiment would be rather more straightforward than measuring/interpreting interfacial water ordering. Therefore, a more relevant question is what kinds of attributes of surface (descriptors based on surface properties) would be more likely to lead to these structural changes of interfacial water in favor of freezing. In other words, if we know enough about the structures and chemistry of a substrate, can we readily predict its freezing ability? I thought that was the grand challenge that motivates this study. As this question is indeed challenging to address, could the authors at least share their thoughts on this?

It is interesting that the dynamics of interfacial water (both mobility and dynamic heterogeneity), which was discussed in homogeneous ice nucleation by the same authors in their PNAS 2019 paper, was not among the most relevant factors for heterogeneous nucleation. I think the authors must have considered this aspect because of their previous study. Does the fact that it's not as relevant as other factors imply there is no appreciable difference in the dynamics of interfacial water induced by different substrates, or the difference cannot be correlated well with water freezing? I think in light of authors' previous study, this point should be commented on, as the results from this study indicate that only the mean structure (and its fluctuation) of interfacial water seems critical.

Regarding the role of density reduction in ice nucleation, although it is probably true that this microscopic principle has not been discussed in the context of heterogeneous ice nucleation, the role of density change has certainly been studied for homogeneous ice nucleation. See for example, Nat. Commun 4, 1887, 2013. The physical basis in enhancing ice nucleation is the same, irrespective of homogeneous or heterogeneous nucleation.

It's also interesting the authors also comment in the end that it is not the most ice-similar structures that are most relevant for ice nucleation, citing that the adsorption uniformity for tetramers and pentamers are more important than hexamers or cages. This can be an important point and in line with some of previous studies showing the role of defects in inducing crystal nucleation. However I was not able to find more details in the main text/Sl. Could this be better discussed?

Reviewer #2 (Remarks to the Author):

The authors have performed a series of simulations to determine the temperature at which ice nucleates on different surfaces. They have also conducted ice adsorption calculations and MD simulations of water on each of these surfaces at the coexistence temperature. From this second set of simulations, they have determined a series of parameters that describe the effect the surface has on the water structure in its vicinity and the interaction between the water and each surface more broadly. The authors have then constructed a model that takes the parameters extracted from these simulations, and that predicts the nucleation temperatures using machine learning. By analyzing this model, they have then been able to determine those features that affect ice nucleation ability.

My view is that this is an elegant piece of work. It does not answer every question about the ice nucleation ability of surfaces, but it produces a set of exciting ideas about this topic that can and should be investigated further. It also provides a simulation toolbox that can be used to tackle other difficult questions of this type via computer simulation. This work should be of interest to other researchers in the field of simulation.

I have chosen not to sign this referee report as I feel I have nothing useful to add and thus should not take any form of credit. The paper clear, and the work is outstanding. All I can do is offer my congratulations on a job well done.

Reviewer #3 (Remarks to the Author):

The authors simulate the nucleation of ice from water on a wide range of crystalline substrates, quantifying the nucleation temperature and use machine-learning method to develop a model for the temperature in terms of a small number of descriptors. These descriptors have physical meaning and therefore help us understand the physics of the nucleation process. The work is very nice, and the presentation is generally very good (with a couple of minor issues I discuss below). It combines an important phenomenon, a novel method, and physical insight. The latter is especially rare in work involving machine learning, which is therefore especially appreciated. I recommend acceptance if my presentation issues are adequately addressed.

I found it quite surprising when I got to the conclusions and discovered that the authors used a coarse-grained model, mW, for water. This is not necessarily a problem, but the conclusion is far too late a place for the authors to state this. They could also use a bit more detail about whether the water-substrate interactions come from. Did they fit any themselves, or are they all from previous publications?

The second thing I thought was missing was a more comprehensive discussion of the descriptors that the authors used. They go into substantial detail about the 4 that were selected as important by their ML process, but I think it is also essential to know, in at least some detail, what other descriptors were available in the pool that the process was selecting from. Some of this information is in the methods section and the SI, but more details (not all details, just more than are there now) need to be added to the main body, since I think it's an essential part of interpreting such fundamental questions as how much meaning to attribute to the final choice of 4 descriptors.

Responses to the Reviewers

For clarity, our responses are highlighted in blue and changes/additions to the manuscript text are highlighted in red.

Reviewer #1 (Remarks to the Author):

This is an interesting modeling study attempting to correlate physical descriptors of a substrate with its heterogeneous ice nucleation efficiency, which is a long-standing scientific question, through machine learning approach. The authors relied on the large database produced by a computationally inexpensive water model and simplified substrate model to enable a data-driven approach for training ML model. The outcome of the study revealed four major descriptors that were shown capable of predicting heterogeneous ice nucleation temperature within a RMSE of $\pm 6\text{K}$. I find this study very well planned and executed, and the results/analysis interesting and insightful.

We thank the reviewer for carefully reading our manuscript and for the supportive assessment.

A main question is that out of the four identified descriptors, only one (lattice match) directly measures surface attributes, whereas other factors describe the structure and energetics of interfacial water induced by surfaces. Although I think it makes perfect physical sense that machine learning model revealed interfacial water are strongly correlated with its likelihood of turning into ice (because in general, the more similar interfacial water is to ice, the more likely it will turn into water as this can be also easily seen from a view of entropy reduction), the outcome of the study appears to benefit modeling study more than experimental study of heterogeneous ice nucleation, since local ordering and density of interfacial water are much easier to characterize in modeling than in experiment--identifying a good/bad nucleator in experiment would be rather more straightforward than measuring/interpreting interfacial water ordering. Therefore, a more relevant question is what kinds of attributes of surface (descriptors based on surface properties) would be more likely to lead to these structural changes of interfacial water in favor of freezing. In other words, if we know enough about the structures and chemistry of a substrate, can we readily predict its freezing ability? I thought that was the grand challenge that motivates this study. As this question is indeed challenging to address, could the authors at least share their thoughts on this?

The reviewer highlights an important point here. We believe that the adsorption energy landscape can also be viewed as surface-property, making 2 of our 4 descriptors directly related to surface chemistry. Furthermore, experimental techniques made tremendous progress in characterizing interfacial water recently, making our results on interfacial water properties interesting for current experiments. A great simplification for this is that we have established that the relevant properties can be obtained at a constant temperature, and not necessarily at or close to T_n .

We agree with the reviewer that a prediction based on simple surface-only properties would be highly desirable. As the reviewer acknowledges, connecting ice nucleation with characteristics of only the substrate is extremely challenging. We believe however, that this work goes some way at achieving this through the following logic. We think it is more feasible

to split the identification of surface characteristics for ice nucleation into two parts: 1) identification of interfacial water characteristics relevant to ice nucleation and 2) identification of pure surface characteristics causing these interfacial water characteristics. This work has focused on point 1) with several suggestions for point 2). We think that this offers new directions for future research. For instance, it is conceivable to perform specific simulation studies trying to correlate substrate chemistry with the interfacial water properties we identified. This however is an effort which requires careful planning and much more work beyond the scope of this study. Furthermore, we are hopeful that through our suggestions also the experimental community will receive inspiration to elucidate aspect 2) mentioned above. We put a few more sentences outlining these thoughts in the discussion:

Two out of the four descriptors we found are more directly related to intrinsic surface properties (lattice match and adsorption energy landscape) while the others describe interfacial water. Overall, it is more desirable to have descriptors that relate to surface chemistry alone. This goal, however, is much more challenging and for future work we propose to split the process of understanding heterogeneous IN into two steps: i) relate IN activity to interfacial water properties and then ii) relate surface chemistry to interfacial water properties.

It is interesting that the dynamics of interfacial water (both mobility and dynamic heterogeneity), which was discussed in homogeneous ice nucleation by the same authors in their PNAS 2019 paper, was not among the most relevant factors for heterogeneous nucleation. I think the authors must have considered this aspect because of their previous study. Does the fact that it's not as relevant as other factors imply there is no appreciable difference in the dynamics of interfacial water induced by different substrates, or the difference cannot be correlated well with water freezing? I think in light of authors' previous study, this point should be commented on, as the results from this study indicate that only the mean structure (and its fluctuation) of interfacial water seems critical.

Indeed, we have considered the dynamics of the liquid layers as well. However, not through the same measurements as was done our 2019 PNAS paper. Characterizing dynamics and dynamical heterogeneity is often non-trivial, the proper length and timescales have to be characterized for each system, which does not seem appropriate for the screening approach taken in this work. Instead, we have opted to characterize the liquid dynamics with more traditional measures of liquid diffusion over many different time scales (both in different layers adjacent to the substrate and distinguishing lateral, perpendicular and overall movements). It is interesting that these have not been discovered to be relevant. In light of reviewers 1 and 3's interest in this topic we have included more discussion on this in the new section dealing with unselected features.

See the very bottom of this document for the new paragraph.

Regarding the role of density reduction in ice nucleation, although it is probably true that this microscopic principle has not been discussed in the context of heterogeneous ice nucleation, the role of density change has certainly been studied for homogeneous ice nucleation. See for example, Nat. Commun 4, 1887, 2013. The physical basis in enhancing ice nucleation is the same, irrespective of homogeneous or heterogeneous nucleation.

While the density reduction in homogeneous nucleation has been noted as suggested by the reviewer, our results show that this i) also applies to heterogeneous nucleation and ii) this means that the density reduction could be used to control the nucleation process. We believe this is an exciting finding with a straightforward physical basis that is only strengthened by

the connection to the homogeneous case. It is even more remarkable since it came out of a purely data-driven approach. We thank the reviewer for pointing this out and make new mention of this connection in the manuscript.

For homogeneous nucleation, Li and coworkers [34] observed a similar trend with the density, presumably for the same physical reason. In addition, our result shows that the density reduction is also a vital component of good heterogeneous nucleators and could thus potentially be used to control heterogeneous nucleation.

It's also interesting the authors also comment in the end that it is not the most ice-similar structures that are most relevant for ice nucleation, citing that the adsorption uniformity for tetramers and pentamers are more important than hexamers or cases. This can be an important point and in line with some of previous studies showing the role of defects in inducing crystal nucleation. However I was not able to find more details in the main text/SI. Could this be better discussed?

We have included more discussion on this aspect, making clearer connections to previous works as pointed out by the reviewer.

The potential role of interfacial structures to the nucleation process has been noted. For instance a defect structure with 5-fold symmetry was observed during homogeneous nucleation [60]. We have not directly observed any particular nucleating structures in this work, but the connection to Eads of pentamers is possible.

Reviewer #2 (Remarks to the Author):

The authors have performed a series of simulations to determine the temperature at which ice nucleates on different surfaces. They have also conducted ice adsorption calculations and MD simulations of water on each of these surfaces at the coexistence temperature. From this second set of simulations, they have determined a series of parameters that describe the effect the surface has on the water structure in its vicinity and the interaction between the water and each surface more broadly. The authors have then constructed a model that takes the parameters extracted from these simulations, and that predicts the nucleation temperatures using machine learning. By analyzing this model, they have then been able to determine those features that affect ice nucleation ability.

My view is that this is an elegant piece of work. It does not answer every question about the ice nucleation ability of surfaces, but it produces a set of exciting ideas about this topic that can and should be investigated further. It also provides a simulation toolbox that can be used to tackle other difficult questions of this type via computer simulation. This work should be of interest to other researchers in the field of simulation.

I have chosen not to sign this referee report as I feel I have nothing useful to add and thus should not take any form of credit. The paper clear, and the work is outstanding. All I can do is offer my congratulations on a job well done.

We are very grateful for the reviewer for assessing our manuscript and providing these encouraging words.

Reviewer #3 (Remarks to the Author):

The authors simulate the nucleation of ice from water on a wide range of crystalline substrates, quantifying the nucleation temperature and use machine-learning method to develop a model for the temperature in terms of a small number of descriptors. These descriptors have physical meaning and therefore help us understand the physics of the nucleation process. The work is very nice, and the presentation is generally very good (with a couple of minor issues I discuss below). It combines an important phenomenon, a novel method, and physical insight. The latter is especially rare in work involving machine learning, which is therefore especially appreciated. I recommend acceptance if my presentation issues are adequately addressed.

We thank the reviewer for their thorough assessment and for interesting suggestions, which we address below.

I found it quite surprising when I got to the conclusions and discovered that the authors used a coarse-grained model, mW, for water. This is not necessarily a problem, but the conclusion is far too late a place for the authors to state this. They could also use a bit more detail about whether the water-substrate interactions come from. Did they fit any themselves, or are they all from previous publications?

It is not our intent to surprise the reader about this aspect and are grateful to the reviewer for pointing out this potential irritation. We have shifted the statement about the model justification to the section before the discussion and hope that this is enough to avoid any surprise or ambiguity about our computational methods.

The second thing I thought was missing was a more comprehensive discussion of the descriptors that the authors used. They go into substantial detail about the 4 that were selected as important by their ML process, but I think it is also essential to know, in at least some detail, what other descriptors were available in the pool that the process was selecting from. Some of this information is in the methods section and the SI, but more details (not all details, just more than are there now) need to be added to the main body, since I think it's an essential part of interpreting such fundamental questions as how much meaning to attribute to the final choice of 4 descriptors.

This is an excellent suggestion. We have followed the reviewers request and added a new section that comes after the discussion of the 4 relevant descriptors. We note that the corpus of not-selected features is much too large to be discussed in its entirety. Thus, we have focused on a few more or less surprising feature types that were not found to be relevant, most notably measures for diffusion and layering. We hope this satisfies the reviewers request and thank them once more or making this important point. The new paragraph reads:

After having discussed the most important descriptor types for predicting T_n , it is interesting to consider the features that have not been selected by our algorithm. First, we note that if a feature has not been selected it does not mean that it is entirely unimportant. This is because our approach takes into consideration both the feature importance and the correlation with already selected features, so a feature not being selected could also come from redundancy with other descriptors. In practice it will be a combination of both effects. Second, the number of non-selected features vastly outnumbers the ones that are selected (for a listing see Supplementary Note 2), so in the following we focus on a few examples that are surprising or are related to previous studies.

We have also computed the layering as defined in [39] and [43,44], where the absolute deviation from bulk number density is measured and integrated, treating increased and decreased densities the same. Our results indicate that rather than an absolute change in density, a density reduction is desirable, hence making a measurement for this the better descriptor. Also, a certain correlation between the layering and density reduction is expected, which might also contribute to layering not being selected.

Most work on heterogeneous nucleation in general focuses on structural aspects of the substrate or the interfacial water. Recent work, however, has highlighted the role of heterogeneous dynamics for supercooled water in the homogeneous case [36]. Despite this, the dynamics near liquid interface remain largely unexplored. To have a simple measure for liquid dynamics near the interface we computed mean displacements at timescales ranging from 1 to 150 ps. A distinction between different layers perpendicular to the surface was also made. None of these features has been selected in the end, which is somewhat of a surprise. A possible explanation is the following: Most of our substrates will decrease the mobility of interfacial water molecules, thus measures of displacements will offer little distinction between them. For hydrophobic surfaces this would not be the case, displacements being substantially higher. However, for those substrates then also trivially other descriptors would indicate the trend with T_n , such as the local ordering, hence making displacements unhelpful.

A large group of features not appearing useful are velocities and forces (distinguished in layers and perpendicular / parallel to the surface normal). These are expected to differ from bulk near the interface, potentially carrying information about T_n . However, this was not the case.